# Error Analysis of a New Five-Degree-of-Freedom Hybrid Robot

Hongjun San *, Lin Ding [ID], Haobin Zhang and Xingmei Wu

Faculty of Mechanical and Electrical Engineering, Kunming University of Science and Technology, Kunming 650504, China; dinglin8227@163.com (L.D.); zhanghaobin202202@163.com (H.Z.); 15923095257@163.com (X.W.)
* Correspondence: sanhjun@163.com

**Abstract:** The error analysis of the robot has a very practical significance for improving its accuracy. Therefore, this paper conducts an error analysis for a new five-degree-of-freedom hybrid robot designed to conduct responsible surface machining. Initially, the error sources of the hybrid robot were sorted out to determine the number of error sources. Then, the error mapping model of the hybrid robot is established by the closed-loop vector method and the first-order perturbation method. Based on the mapping property of the 6th-order velocity Jacobi matrix, the compensable and non-compensable error sources affecting the posture error at the end of the hybrid robot are separated. Finally, the error analysis of the separated error sources is carried out to study the effect of single error sources and multiple error sources coupled with the posture error at the end of the robot. The results show that among the individual error sources, the dynamic and fixed platform hinge position error has the most significant effect on the end of the robot; among the integrated posture errors after coupling multiple error sources, the position of the dynamic and fixed platform hinge position error and the translational joint initial position dominate; the analysis of the different trajectories also yields that the error introduced by each error source increases gradually with the increase of the end trajectory. When designing this hybrid robot, attention should be paid to the manufacturing and installation accuracy of the dynamic and fixed platform hinge point positions and the translational joint initial position.

**Keywords:** error sources; error modeling; error separation; error analysis; error coupling

## 1. Introduction

In this paper, a new five-degree-of-freedom hybrid robot is designed. The robot can be used for machining surfaces, and its main body (3PRS parallel mechanism) is the key component responsible for machining. However, in the manufacturing, assembly, and use of parallel robots, under the interference of control system accuracy, applied load, and the external environment [1–3], there will exist errors between the actual and nominal positions of the robot end, and these errors will reduce the positioning accuracy at the end of the parallel mechanism [4,5]. Methods to improve robot positioning accuracy include error prevention and parameter calibration [6–8], and error analysis of the mechanism is the basis for both of these methods [9]. Based on the error analysis, the key factors that generate errors in the mechanism can be clarified, which can guide the manufacturing assembly of the hybrid robot.

To perform an error analysis of the mechanism, we must first identify the sources of error in the mechanism. Many factors influence the error in robot end posture accuracy, and the error sources can be categorized into different types according to the different formation causes. According to the different time characteristics, the error sources can be divided into static error and dynamic error [10]. Static errors are caused by geometric parameter errors, machining errors, and assembly errors, and dynamic errors are caused by vibration deformation and wear of the robot. Studies have found that static errors are the main source of errors in robots, which can account for up to about 80% [11]. Li et al. [12],

based on a designed truss hybrid casting robot, investigated the effect of error sources on the robot's end position by constructing an error transfer matrix and concluded that machining error is the main source of error. Ye et al. [13] studied the geometric error of a five-degree-of-freedom hybrid robot and reduced the positional error at the end of the robot by about 90% by kinematic calibration. Shen et al. [7] studied the effect of geometric parameters on the robot end, and the absolute positioning accuracy of the robot end was improved by sensitivity analysis. Therefore, in this paper, we will mainly study the effect of static error on the robot's end position error and disregard the effect of other factors for the time being.

Based on the error source, constructing a mapping relationship between the error source and the end-position error is a key step in the error analysis of the mechanism. The main modeling ideas for parallel and hybrid robots are the closed-loop vector method [14,15] and the branched-chain analysis method. Branched-chain analysis methods include the D-H matrix method [16,17] and the spinor method [18]. Shan and Cheng [19], based on the vector method in the first-order perturbation condition, established an error mapping model for a 2(3PUS+S) parallel robot that contains machining error, assembly error, and ball-joint clearance. Finally, we calibrated the robot, and the robot's end-positioning error was significantly reduced after the calibration. Zhang et al. [20], based on the D-H matrix method, established a geometric error model for 2(3HUS+S) parallel robots and carried out calibration experiments to verify the correctness of the error model establishment. Huang et al. [21], based on screw theory, established an error model for a six-degree-of-freedom hybrid robot that included all error sources of joints and linkages and compensated the robot for the errors so that the robot gained satisfactory positional accuracy in the workspace. The error model established by the closed-loop vector equation perturbation analysis method can include all structural errors and avoid differential operations, and the derivation process is simple and applicable to a wide range. The D-H matrix method does not allow for error modeling of the mechanism using a uniform mathematical expression and is computationally complex. By using the spinor method to establish the error model, a unified characterization of the error model can be obtained, but due to the transient nature of the spinor, it is necessary to use other mathematical methods for the position analysis of the mechanism, so the mechanism analysis is complicated. Therefore, in this paper, the error modeling of the robot is performed using the closed-loop vector equation perturbation method.

In the hybrid robot, the 3PRS parallel mechanism is a parallel mechanism with few degrees of freedom, and there are compensable and non-compensable errors in the end position errors [22]. The compensable errors can be fully compensated using kinematic calibration, and the non-compensable errors need to be strictly controlled in the manufacturing assembly. So, it is necessary to separate the compensable and non-compensable error sources in the error model. Huang et al. [23–26] investigated a series of two-, three-, and four-degree-of-freedom error modeling methods based on the closed-loop vector method containing parallelogram support chains with few degrees of freedom and separated the compensable and non-compensable errors affecting the end-position error. Liu et al. [27] proposed a generalized error modeling method with fewer degrees of freedom that can effectively separate compensable and non-compensable error sources and guide the improvement of accuracy in manufacturing and assembly processes. In most of the existing studies, the effects of individual error sources are studied on the separated error sources [14,16,17,28], and the coupling characteristics of each error source are less studied. Therefore, in this paper, the effects of individual error sources and the integrated position error after coupling multiple error sources are studied based on the established error model and the separated error sources. By considering the influence of a single error source and multiple error sources coupled with the robot end-position error, the error sources that have a significant influence on the robot end-position error are identified.

In this paper, an error analysis study is conducted for a new five-degree-of-freedom hybrid robot. First, the error sources of the hybrid robot were traced, and the traceability

process defined the source and the amount of each error. The kinematic analysis of the parallel mechanism of the hybrid robot was carried out by the vector equation method, and the Jacobi matrix was established. Then the closed-loop vector perturbation method is introduced to establish the mapping relationship between the geometric error source of the robot and the end-position error. By mapping the properties of the Jacobi matrix, the compensable and non-compensable error sources affecting the end position error are separated. Finally, the error analysis is performed on the isolated error sources to study the influence of individual error sources on the end position error of the robot and the influence of the integrated position error on the end position error of the robot after coupling multiple error sources. The simulation results show that the established error model is correct and that significant factors affecting the robot end position error can be found.

## 2. Virtual Prototype Model Description

The structure of the new five-degree-of-freedom hybrid robot is shown in Figure 1. The new hybrid robot consists of a tandem mechanism composed of a set of slide rails and a 3PRS parallel mechanism. This robot is capable of machining surfaces and is a single-point operation hybrid robot. This hybrid robot combines the features of a large working space and easy control of the tandem mechanism with structural stability; high stiffness, and low error accumulation of the parallel mechanism, so the use of the hybrid mechanism can improve the machining accuracy of the robot. The new five-degree-of-freedom hybrid robot can be noted as $B + SKM^2 + PKM^3$ (B representative rack), and the orientation feature set can be expressed as $M_{F^5}^1 \left( F^5 = j_1^d \left( SKM^2 + PKM^3 \right) \right)$. Among them, $M_{F^5}^1$ represents a robot with a total number of degrees of freedom of 5 and only one operation chain, $j_1^d$ represents the number of 5 degrees of freedom of the output member of this operation chain, which is generated by a tandem mechanism with 2 degrees of freedom (denoted as $SKM^2$) and a parallel mechanism with 3 degrees of freedom (denoted as $PKM^3$) [29].

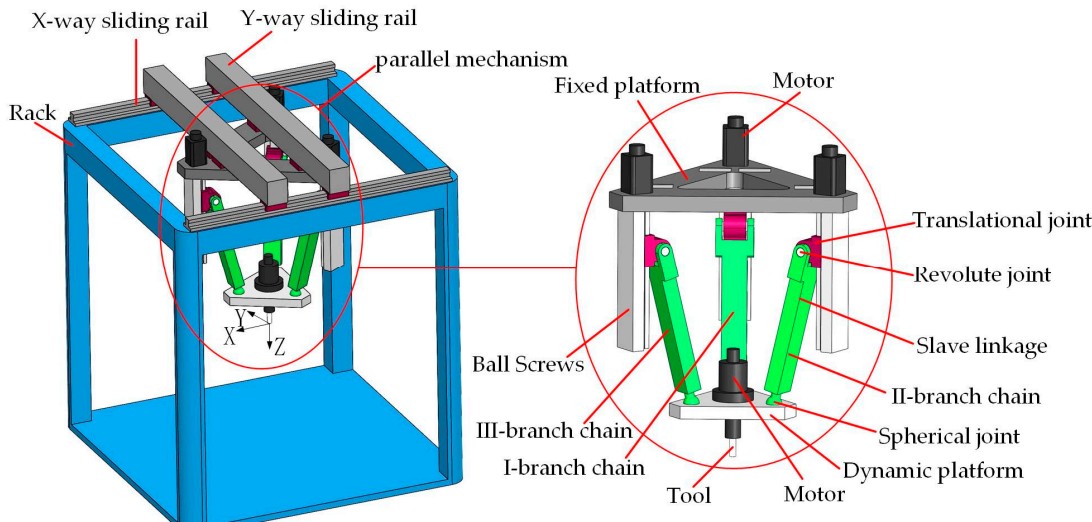

**Figure 1.** New five degrees of freedom hybrid robot 3D drawing.

The XY tandem mechanism is responsible for adjusting the position of the end in the XY plane, which consists of two nominally orthogonal sliding subsets. The 3PRS parallel mechanism consists of a fixed platform, a dynamic platform, and three identical support chains between the two platforms. It is capable of rotating the end around the X-axis and the Y-axis and moving up and down along the Z-axis. Each PRS chain contains a translation joint, a revolute joint, a slave linkage, and a spherical joint. The fixed platform and the dynamic platform are both equilateral triangles. The fixed platform has ball screws fixedly attached at three corners, and the translational joint is connected to the ball screws. One

end of the slave linkage is connected to the translational joint through a revolute joint, and the other end of the slave linkage is connected to the dynamic platform through a spherical joint. The translational joint on each of the supporting chains is a driving sub, which drives the dynamic platform to move in space relative to the fixed platform. A tool is attached to the center of the dynamic platform and can be rotated in the vertical direction.

### 3. Geometric Error Source Analysis

For establishing the error model of a hybrid robot, find out the most significant factors that affect the end position error of the robot. It is necessary to analyze the mechanism composition principle and structural characteristics of the robot in detail and thus determine the source of geometric errors in the hybrid robot. Compared with the two-degree-of-freedom tandem slide, the 3PRS parallel mechanism's geometric error source is the primary factor affecting the end posture error of the hybrid robot, so this paper focuses on analyzing the parallel mechanism's error problem.

Initially, establish the coordinate system of the tandem part of the hybrid robot as shown in Figure 2. The coordinate system $\{O\} = \{O : X, Y, Z\}$ is the fixed coordinate system of the rack, the origin $O$ is the geometric orthogonal center of the plane on the frame, the X-axis points from the origin $O$ to $A_1$, the Y-axis is parallel to $A_2A_3$, the direction from $A_3$ to $A_2$, the Z-axis satisfies the right-hand rule, and the direction is downward. $OA_1$ is parallel to the X-way slide and $A_2A_3$ is parallel to the Y-way slide. The coordinate system $\{O\}$ is the calibration coordinate system of the hybrid robot.

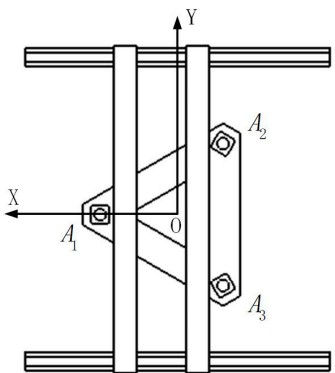

**Figure 2.** Tandem slide coordinate system.

The hybrid robot's parallel mechanism part of the structure sketch is shown in Figure 3. The coordinate system $\{O\}$ in the figure and the coordinate system $\{O\}$ in the tandem slide are the same coordinate system. Fixed platform transition coordinate system $\{O_i\} = \{O_i : x_i, y_i, z_i\}(i = 1, 2, 3)$ for the system $\{O\}$ around the Z-axis rotation $\alpha_i$ formation, among them $\alpha_i = \frac{2(i-1)\pi}{3}$. Fixed platform hinge point connected coordinate system $\{A_i\} = \{A_i : x_{A_i}, y_{A_i}, z_{A_i}\}(i = 1, 2, 3)$, its origin $A_i$ is the center of the fixed platform articulation point, $x_{A_i}$ along the $OA_i$ direction and perpendicular to the ball screw guide, $z_{A_i}$ and the direction of the guide overlap, the direction down, $y_{A_i}$ meet the right-hand rule. Let the vector $a_i$ be the nominal vector from $\{A_i\}$ to $\{O_i\}$, and $\Delta a_i$ and $\theta_{A_i}$ be the position error vector and attitude error vector of the coordinate system $\{A_i\}$ relative to the coordinate system $\{O\}$, respectively. $a_i$, $\Delta a_i$, and $\theta_{A_i}$ are measured under the coordinate system $\{O\}$. The translation joint connected coordinate system $\{B_i\} = \{B_i : x_{B_i}, y_{B_i}, z_{B_i}\}(i = 1, 2, 3)$, whose origin $B_i$ is the intersection of the translation joint axis and the revolute joint theoretical axis; $y_{B_i}$ is along the direction of the revolute joint theoretical axis, pointing to the same $y_{A_i}$; $z_{B_i}$ is along the direction of the translation joint rail movement; direction downward; and $x_{B_i}$ satisfies the right-hand rule. Define $\Delta q_i$ as the initial position error of the coordinate system $\{B_i\}$, $\theta_{B_i}$ as the attitude error vector of the coordinate system $\{B_i\}$ with respect to the coordinate system $\{A_i\}$, where the perpendicularity error of the translation joint can be expressed by $\theta_{B_i}$. Both $\Delta q_i$ and $\theta_{B_i}$

are measured under the coordinate system $\{A_i\}$. The gap error figure of the revolute joint is shown in Figure 4. In the figure of the connected coordinate system of a slave linkage revolute joint $\{B'_i\} = \left\{ B'_i : x_{B'_i}, y_{B'_i}, z_{B'_i} \right\} (i = 1, 2, 3)$, the coordinate system $\{B'_i\}$ is formed by the rotation of the coordinate system $\{B_i\}$ around $y_{B_i}$ and the angle $\theta_i$; its origin $B'_i$ is the intersection of the actual axis of the revolute joint-linkage and the axis of the translation joint-linkage; $y_{B'_i}$ and the actual axis of the revolute joint-linkage overlap, pointing to the same $y_{B_i}$; $z_{B'_i}$ along the direction of the slave linkage, pointing to $C_i$; and $x_{B'_i}$ satisfies the right-hand rule. The vectors $\Delta b_i$ and $\theta_{B'_i}$ are the position error vector and attitude error vector of the coordinate system $\{B'_i\}$ relative to the coordinate system $\{B_i\}$, respectively. The swing angle error of the slave linkage can be expressed by $\theta_{B'_i}$, and both $\Delta b_i$ and $\theta_{B'_i}$ are measured under the coordinate system $\{B_i\}$.

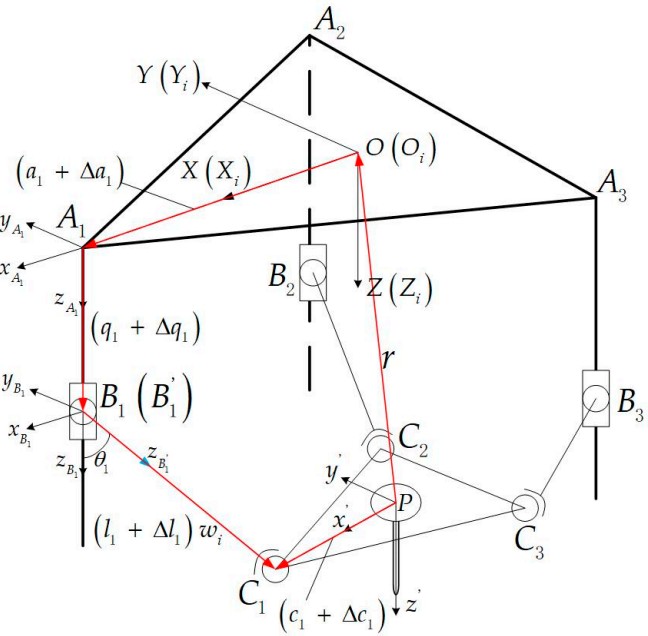

**Figure 3.** 3-PRS parallel mechanism sketch.

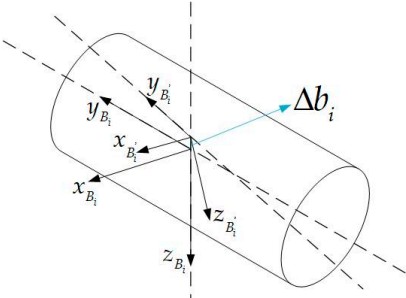

**Figure 4.** Revolute joint coordinate system.

The dynamic platform connection coordinate system of the parallel mechanism of the hybrid robot (the end connection coordinate system of the hybrid robot) $\{P\} = \{P : x', y', z'\}$ is parallel to the coordinate system $\{O\}$, and the attitude is the same as the coordinate system $\{O\}$. The dynamic platform transition coordinate system $\{P'_i\} = \left\{ P'_i : x_{P'_i}, y_{P'_i}, z_{P'_i} \right\} (i = 1, 2, 3)$ is formed by the rotation $\alpha_i$ of the system $\{P\}$ around the $Z'$-axis. Define $r$ and $\Delta r$ as the nominal vector and error vector of point $P$ relative to point $O$, respectively. The attitude error vector of coordinate system $\{P\}$ relative to coordinate system $\{O\}$ as $\theta$. $r$, $\Delta r$, and $\theta$ are measured under a coordinate system $\{O\}$. Note that

$C_i$ is the center of the hinge position of the spherical joint and the dynamic platform, the vector $c_i$ is the nominal vector from point $C_i$ to point $P$, and $\Delta c_i$ is the position error vector $C_i$ relative to point $P$.

So far, all the errors in the parallel mechanism part of the hybrid robot have been shown. Including the position and attitude errors of the fixed platform hinge point, the errors of the initial position of the translation joint, the perpendicularity errors of the translation joint, the errors of the revolute joint gap, the errors of the swing angle of the slave linkage, and the errors of the position of the dynamic platform hinge point.

## 4. Error Modeling

Error modeling refers to the establishment of a functional mapping relationship between the geometric error source and the end-posture error. Since the geometric error source of the 3PRS parallel mechanism is the main factor affecting the end position error of the hybrid robot, this paper systematically studies the geometric error model of the 3PRS parallel mechanism. The 3PRS parallel mechanism belongs to the less-degree-of-freedom parallel mechanism, and its end posture errors include compensable and non-compensable errors. The compensable error can be fully compensated using kinematic calibration, while the non-compensable error needs to be strictly controlled during manufacturing and assembly.

The error mapping model of the parallel mechanism is first developed by using the closed-loop vector method and the first-order perturbation method. Then the error sources of the parallel mechanism are separated by the mapping property of the 6th-order velocity Jacobi matrix. Sixth-order velocity Jacobi matrices are mapped in such a way that the sources of error mapped to the drive space are compensable sources of error, and the sources of error mapped to the constraint space are non-compensable sources of error [27].

*4.1. Kinematic Analysis of Parallel Mechanisms*

4.1.1. Establishment of Inverse Kinematic Model for the Parallel Mechanism

In the 3-PRS parallel mechanism, due to the presence of the constraining effect of the revolute joint, its point $C_i$ can only move within the three constraining surfaces. These three constraining surfaces are the three surfaces where the direction of movement of the translational joint is tensioned with the slave linkage. This constraint relationship can be derived as a function of the three constrained positional variables $(x, y, \gamma)$ and the independent positional variables $(\alpha, \beta, z)$.

The rotation matrix of the dynamic platform coordinate system concerning the fixed coordinate system $\{O\}$ is $R$. Using the $X - Y - Z$ Euler angle description, it can be expressed as

$$R = \begin{bmatrix} \cos\beta\cos\gamma & -\cos\beta\sin\gamma & \sin\beta \\ \sin\alpha\sin\beta\cos\gamma + \cos\alpha\sin\gamma & -\sin\alpha\sin\beta\sin\gamma + \cos\alpha\cos\gamma & -\sin\alpha\cos\beta \\ -\cos\alpha\sin\beta\cos\gamma + \sin\alpha\sin\gamma & \cos\alpha\sin\beta\sin\gamma + \sin\alpha\cos\gamma & \cos\alpha\cos\beta \end{bmatrix}. \tag{1}$$

The position vector coordinates of the point $B_i$ in the fixed coordinate system $\{O\}$ can be expressed as

$$\begin{cases} {}^oB_1 = \begin{bmatrix} {}^oB_{1x} & {}^oB_{1y} & {}^oB_{1z} \end{bmatrix}^T = \begin{bmatrix} a & 0 & q_1 \end{bmatrix}^T \\ {}^oB_2 = \begin{bmatrix} {}^oB_{2x} & {}^oB_{2y} & {}^oB_{2z} \end{bmatrix}^T = \begin{bmatrix} -\frac{1}{2}a & \frac{\sqrt{3}}{2}a & q_2 \end{bmatrix}^T \\ {}^oB_3 = \begin{bmatrix} {}^oB_{3x} & {}^oB_{3y} & {}^oB_{3z} \end{bmatrix}^T = \begin{bmatrix} -\frac{1}{2}a & -\frac{\sqrt{3}}{2}a & q_2 \end{bmatrix}^T \end{cases} ; \tag{2}$$

among them: $a$ represents the radius of the inscribed circle of the fixed platform and $q_i$ represents the distance traveled by the translational joint along the guide rail.

The position vector coordinates of the point $C_i$ in the dynamic coordinate system $\{P\}$ can be expressed as

$$
\begin{cases}
{}^P C_1 = \begin{bmatrix} {}^P C_{1x} & {}^P C_{1y} & {}^P C_{1z} \end{bmatrix}^T = \begin{bmatrix} c & 0 & 0 \end{bmatrix}^T \\
{}^P C_2 = \begin{bmatrix} {}^P C_{2x} & {}^P C_{2y} & {}^P C_{2z} \end{bmatrix}^T = \begin{bmatrix} -\frac{1}{2}c & \frac{\sqrt{3}}{2}c & 0 \end{bmatrix}^T \\
{}^P C_3 = \begin{bmatrix} {}^P C_{3x} & {}^P C_{3y} & {}^P C_{3z} \end{bmatrix}^T = \begin{bmatrix} -\frac{1}{2}c & -\frac{\sqrt{3}}{2}c & 0 \end{bmatrix}^T
\end{cases} ;
\tag{3}
$$

among them: $c$ represents the radius of the inscribed circle of the moving platform.

The position vector of the point $P$ in the fixed coordinate system $\{O\}$ is $r = \begin{bmatrix} x & y & z \end{bmatrix}^T$.

In the kinematic analysis, each vector is represented in the fixed coordinate system $\{O\}$. The position vector of the point $C_i$ in the fixed coordinate system $\{O\}$ can be expressed as

$$
{}^O C_i = \begin{bmatrix} {}^O C_{ix} & {}^O C_{iy} & {}^O C_{iz} \end{bmatrix}^T = R {}^P C_i + r = \begin{bmatrix} R_{11}{}^P C_{ix} + R_{12}{}^P C_{iy} + R_{13}{}^P C_{iz} + x \\ R_{21}{}^P C_{ix} + R_{22}{}^P C_{iy} + R_{23}{}^P C_{iz} + y \\ R_{31}{}^P C_{ix} + R_{32}{}^P C_{iy} + R_{33}{}^P C_{iz} + z \end{bmatrix} ;
\tag{4}
$$

among them: $R_{ij}$ represents the element in the row $i$ and column $j$ of the corresponding rotation matrix.

Ideally, the three constraint surfaces of the parallel mechanism become $120°$ uniformly distributed about the center of the dynamic platform, and the constraint equation of the three constraint surfaces can be obtained as

$$
\begin{cases}
{}^O C_{1y} = 0 \\
{}^O C_{2y} = -\sqrt{3}{}^O C_{2x} \\
{}^O C_{3y} = \sqrt{3}{}^O C_{3x}
\end{cases} .
\tag{5}
$$

Substituting Equations (1), (3), and (4) into Equation (5) can be simplified to obtain the expression of the functional relationship between the constrained variable $(x, y, \gamma)$ and the independent variable $(\alpha, \beta, z)$ as

$$
\begin{aligned}
x &= \tfrac{1}{2}c(\cos\beta\cos\gamma + \sin\alpha\sin\beta\sin\gamma - \cos\alpha\cos\gamma) \\
y &= -c(\sin\alpha\sin\beta\cos\gamma + \cos\alpha\sin\gamma) \\
\gamma &= -\arctan\left(\frac{\sin\alpha\sin\beta}{\cos\alpha + \cos\beta}\right)
\end{aligned} .
\tag{6}
$$

To solve the position inverse solution of the parallel mechanism, i.e., the known position vector coordinates of the end point P, inversely solve for the displacement $q_i$ of the driving sub. The closed-loop vector method is utilized here to solve the inverse solution of the parallel mechanism through the slave linkage rod length constraint.

The vector $l_i$ from the slave linkage in the fixed coordinate system $\{O\}$, according to the triangular vector rule, yields Equation (7).

$$
l_i = {}^O C_i - {}^O B_i, (i = 1, 2, 3).
\tag{7}
$$

Since the slave linkage length is constant, it has

$$
|l_i| \equiv l, (i = 1, 2, 3).
\tag{8}
$$

Bringing Equation (7) into Equation (8) to solve the system of equations can obtain the displacement $q_i$ of the driving sub of the parallel mechanism. The inverse solution of the parallel mechanism can be obtained as follows:

$$
\begin{cases}
q_1 = cR_{31} + z - \sqrt{l2 - (cR_{11} + x - a)^2} \\
q_2 = -\frac{1}{2}cR_{31} + \frac{\sqrt{3}}{2}cR_{32} + z \\
\quad - \sqrt{l2 - \left(-\frac{1}{2}cR_{11} + \frac{\sqrt{3}}{2}cR_{12} + x + \frac{1}{2}a\right)^2 - \left(-\frac{1}{2}cR_{21} + \frac{\sqrt{3}}{2}cR_{22} + y - \frac{\sqrt{3}}{2}a\right)^2} \\
q_3 = -\frac{1}{2}cR_{31} - \frac{\sqrt{3}}{2}cR_{32} + z \\
\quad - \sqrt{l2 - \left(-\frac{1}{2}cR_{11} - \frac{\sqrt{3}}{2}cR_{12} + x + \frac{1}{2}a\right)^2 - \left(-\frac{1}{2}cR_{21} - \frac{\sqrt{3}}{2}cR_{22} + y + \frac{\sqrt{3}}{2}a\right)^2}
\end{cases}.
$$

### 4.1.2. Establishment of 6th Order Velocity Jacobi Matrix for the Parallel Mechanism

Use the direct derivation method of the displacement equation to solve the Jacobi matrix of the parallel mechanism. From the vector relationship in the sketch of the parallel mechanism shown in Figure 3, choosing one of the branch chains (here the red closed-loop vector chain in Figure 3 is used as an example), one gets:

$$
\overrightarrow{PO} + \overrightarrow{PC_i} = \overrightarrow{OA_i} + \overrightarrow{A_iB_i} + \overrightarrow{B_iC_i}
$$

can also be written as a vector equation:

$$
\boldsymbol{r} + \boldsymbol{c_i} = \boldsymbol{a_i} + q_i\boldsymbol{Z_{A_i}} + l_i\boldsymbol{w_i}, \tag{9}
$$

among them: $\boldsymbol{Z_{A_i}}$ represents the unit vector of $A_iB_i$ in branched chain $i$ and $\boldsymbol{w_i}$ represents the unit vector of fixed-length link in the branched chain $i$. $\boldsymbol{a_i} = \boldsymbol{R_i}\boldsymbol{a_{io}}$, $\boldsymbol{c_i} = \boldsymbol{RR_i}\boldsymbol{c_{i0}}$, $\boldsymbol{Z_{A_i}} = \boldsymbol{e_3} = \begin{pmatrix} 0 & 0 & 1 \end{pmatrix}^T$, and $\boldsymbol{w_i} = \boldsymbol{R_i}\boldsymbol{R_{Bi}}\boldsymbol{e_3}$.

The rotation matrix of fixed platform transition coordinate system $\{O_i\}$ with respect to coordinate system $\{O\}$ is $\boldsymbol{R_i}$; the rotation matrix of coordinate system $\{B_i'\}$ with respect to coordinate system $\{B_i\}$ is $\boldsymbol{R_{B_i}}$.

$$
\boldsymbol{R_i} = \begin{bmatrix} \cos\alpha_i & -\sin\alpha_i & 0 \\ \sin\alpha_i & \cos\alpha_i & 0 \\ 0 & 0 & 1 \end{bmatrix}, \quad \boldsymbol{R_{B_i}} = \begin{bmatrix} \cos\theta_i & 0 & \sin\theta_i \\ 0 & 1 & 0 \\ -\sin\theta_i & 0 & \cos\theta_i \end{bmatrix}.
$$

Derive the nominal vector equation of Equation (9) concerning time. The velocity relationship expression (10) is obtained for the parallel mechanism:

$$
\dot{\boldsymbol{r}} + \boldsymbol{\omega} \times \boldsymbol{c_i} = \dot{q}_i\boldsymbol{e_3} + \boldsymbol{\omega_i} \times l_i\boldsymbol{w_i}; \tag{10}
$$

among them: $\dot{\boldsymbol{r}}$: linear velocity of the reference point $P$ of the moving platform, $\boldsymbol{\omega}$: angular velocity of the moving platform, $\dot{q}_i$: drive rate of the drive joint in the branch chain $i$, and $\boldsymbol{\omega_i}$: angular velocity of the fixed-length connecting rod in the branch chain $i$.

The left point multiplication $w_i$ at both ends of the velocity relation (10). Projecting to the space of motion yields the velocity drive equation for the parallel mechanism (11):

$$
\boldsymbol{w_i^T}\left(\dot{\boldsymbol{r}} + \boldsymbol{\omega} \times \boldsymbol{c_i}\right) = \dot{q}_iw_{iz}; \tag{11}
$$

among them: $w_{iz} = \boldsymbol{w_i^T}\boldsymbol{e_3}$, $\boldsymbol{w_i^T}(\boldsymbol{\omega_i} \times \boldsymbol{w_i}) = 0$.

Simplify Equation (11) as:

$$
w_{iz}^{-1}(\boldsymbol{c_i} \times \boldsymbol{w_i})^T\boldsymbol{\omega} + w_{iz}^{-1}\boldsymbol{w_i^T}\dot{\boldsymbol{r}} = \dot{q}_i. \tag{12}
$$

Since the slave linkage is constrained by the revolute, the three slave linkages can only move in the plane of their revolute constraint. The left point multiplication $v_i(v_i = R_ie_2)$ of

the velocity relation (10). The projection to the constraint space yields the velocity constraint Equation (13) for the parallel mechanism:

$$(c_i \times v_i)^T \omega + v_i^T \dot{r} = 0. \tag{13}$$

Combining Equations (12) and (13) is written in matrix form to obtain the speed mapping equation for the parallel mechanism:

$$\dot{q}_i = JM;$$

among them:

$$\dot{q}_i = \begin{pmatrix} \dot{q}_1 & \dot{q}_2 & \dot{q}_3 & \vdots & 0 & 0 & 0 \end{pmatrix}^T, \ J = \begin{bmatrix} J_a \\ \cdots \\ J_c \end{bmatrix} = \begin{bmatrix} w_{1z}^{-1}(c_1 \times w_1)^T & w_{1z}^{-1}w_1^T \\ w_{2z}^{-1}(c_2 \times w_2)^T & w_{2z}^{-1}w_2^T \\ w_{3z}^{-1}(c_3 \times w_3)^T & w_{3z}^{-1}w_3^T \\ \cdots & \cdots \\ (c_1 \times v_1)^T & v_1^T \\ (c_2 \times v_2)^T & v_2^T \\ (c_3 \times v_3)^T & v_3^T \end{bmatrix}_{(6 \times 6)},$$

$$M = \begin{bmatrix} \omega \\ \dot{r} \end{bmatrix};$$

among them: $J$ is the 6th order inverse velocity Jacobi matrix, also known as velocity Jacobi matrix, $J_a$ is the driving Jacobi matrix, and $J_c$ is the constrained Jacobi matrix.

### 4.2. Error Modeling of 3PRS Parallel Mechanism

Based on the above definition of the coordinate system and geometric error sources, the first-order perturbation of the nominal vector Equation (9), ignoring the higher-order terms, yields the vector Equation (14) containing the error:

$$\begin{aligned} &r + \Delta r + [E_3 + \theta\times]RR_i(c_{i0} + \Delta c_{i0}) = R_i(a_{i0} + \Delta a_{i0}) + R_i([E_3 + \theta_{Ai}\times](q_i + \Delta q_i)e_3) \\ &+R_i([E_3 + \theta_{Ai}\times][E_3 + \theta_{Bi}\times]\Delta b_i) + R_i\left([E_3 + \theta_{Ai}\times][E_3 + \theta_{Bi}\times]R_{Bi}\Big[E_3 + \theta_{B_i'}\times\Big](l + \Delta l_i)e_3\right) \end{aligned} ; \tag{14}$$

among them: $E_3$: Third order unit matrix; $\theta\times$: Antisymmetric matrix of the attitude error vector $\theta$, a first-order tensor, can do the fork multiplication operation, $\boldsymbol{\theta} = \begin{pmatrix} \theta_x & \theta_y & \theta_z \end{pmatrix}^T$ then $\theta\times = \begin{bmatrix} 0 & -\theta_z & \theta_y \\ \theta_z & 0 & -\theta_x \\ -\theta_y & \theta_x & 0 \end{bmatrix}$.

Subtracting the nominal vector Equation (9) from the vector Equation (14) containing the error and retaining the linear term yields:

$$\begin{aligned} &\theta \times c_i + \Delta r = \Delta a_i + R_i\Delta q_i e_3 + R_i(\theta_{Ai} \times q_i e_3) + R_i\Delta b_i + \Delta l_i w_i + l_i(R_i\theta_{A_i}) \times w_i + l_i(R_i\theta_{B_i}) \times w_i \\ &+l_i\left(R_iR_{B_i}\theta_{B_i}'\right) \times w_i - \Delta c_i \end{aligned}, \tag{15}$$

among them: $\Delta a_i = R_i\Delta a_{i0}$, $\Delta c_i = RR_i\Delta c_{i0}$, $w_i = R_iR_{B_i}e_3$.

The left point multiplication $w_i$ at both ends of Equation (15). Projecting to the motion space yields:

$$(c_i \times w_i)^T \theta + w_i^T \Delta r = w_i^T \Delta a_i + w_i^T R_i \Delta q_i e_3 + w_i^T R_i(\theta_{Ai} \times q_i e_3) + w_i^T R_i \Delta b_i + \Delta l_i - w_i^T \Delta c_i; \tag{16}$$

among them: $w_i^T l_i(R_i\theta_{A_i}) \times w_i = w_i^T l_i(R_i\theta_{B_i}) \times w_i = w_i^T l_i\left(R_iR_{B_i}\theta_{B_i}'\right) \times w_i = 0$.

The left point multiplication $v_i$ at both ends of Equation (15). Projecting to the constraint space yields:

$$(c_i \times v_i)^T \theta + v_i^T \Delta r = v_i^T \Delta a_i + v_i^T R_i(\theta_{Ai} \times q_i e_3) + v_i^T R_i \Delta b_i + v_i^T l_i(R_i \theta_{A_i}) \times w_i + v_i^T l_i(R_i \theta_{B_i}) \times w_i$$
$$+ v_i^T l_i \left( R_i R_{B_i} \theta_{B'_i} \right) \times w_i - v_i^T \Delta c_i \quad ; \tag{17}$$

among them: $v_i^T R_i \Delta q_i e_3 = v_i^T \Delta l_i w_i = 0$.

Combining vertical (16) and Equation (17) is written in matrix form to obtain the error model Equation (18) for the parallel mechanism:

$$J\Delta = \begin{bmatrix} A & 0 \\ 0 & B \end{bmatrix} \varepsilon, \tag{18}$$

among them: $J = \begin{bmatrix} J_a & J_c \end{bmatrix}^T$, $\Delta = \begin{bmatrix} \theta \\ \Delta r \end{bmatrix}$, $A = \begin{bmatrix} a_{11} & 0 & 0 \\ 0 & a_{22} & 0 \\ 0 & 0 & a_{33} \end{bmatrix}$, $B = \begin{bmatrix} b_{11} & 0 & 0 \\ 0 & b_{22} & 0 \\ 0 & 0 & b_{33} \end{bmatrix}$,

$$J_a = \begin{bmatrix} w_{1z}^{-1}(c_1 \times w_1)^T & w_{1z}^{-1} w_1^T \\ w_{2z}^{-1}(c_2 \times w_2)^T & w_{2z}^{-1} w_2^T \\ w_{3z}^{-1}(c_3 \times w_3)^T & w_{3z}^{-1} w_3^T \end{bmatrix}, \quad J_c = \begin{bmatrix} (c_1 \times v_1)^T & v_1^T \\ (c_2 \times v_2)^T & v_2^T \\ (c_3 \times v_3)^T & v_3^T \end{bmatrix},$$

$a_{ii} = \begin{bmatrix} w_{iz}^{-1}\sin\theta_i & w_{iz}^{-1}\cos\theta_i & w_{iz}^{-1}\cos\theta_i & q_i w_{iz}^{-1}\sin\theta_i & w_{iz}^{-1}\sin\theta_i & w_{iz}^{-1}\cos\theta_i & w_{iz}^{-1} & -w_{iz}^{-1}w_i^T R R_i \end{bmatrix}$

$b_{ii} = \begin{bmatrix} 1 & -q_i - l_i\cos\theta_i & l_i\sin\theta_i & 1 & -l_i\cos\theta_i & l_i\sin\theta_i & -1 \end{bmatrix}$,

$\varepsilon = \begin{bmatrix} \varepsilon_a & \varepsilon_c \end{bmatrix}^T$,

$\varepsilon_a = \begin{bmatrix} \Delta a_{iox} & \Delta a_{ioz} & \Delta q_i & \theta_{Aiy} & \Delta b_{ix} & \Delta b_{iz} & \Delta l_i & \Delta c_{io} \end{bmatrix}^T$, and

$\varepsilon_c = \begin{bmatrix} \Delta a_{ioy} & \theta_{A_{ix}} & \theta_{A_{iz}} & \Delta b_{iy} & \theta_{B_{ix}} & \theta_{B_{iz}} & \theta_{B'_{ix}} \end{bmatrix}$.

Error model Equation (18) which $J$ is the 6th order velocity Jacobi matrix, $J_a$ is the driving Jacobi matrix, and $J_c$ is the constrained Jacobi matrix. The geometric error source projected to the motion space is the compensable error, and the geometric error source projected to the constrained space is the non-compensable error. According to the error model equation, it is known that $\varepsilon_a$ is a compensable error source with 24 terms. Compensable error sources are the components $\Delta a_{iox}$, $\Delta a_{ioz}$ of the fixed platform hinge point position error $\Delta a_{io}$ on the $x_i$ axis and $z_i$ axis in the system $\{O\}$; translation joint initial position error $\Delta q_i$; the component $\theta_{A_{iy}}$ of the attitude error vector $\theta_{A_i}$ on the $y_i$-axis of the system $\{O_i\}$; the components $\Delta b_{ix}$, $\Delta b_{iz}$ of the revolute position error vector $\Delta b_i$ on the $x_{B_i}$ and $z_{B_i}$ axes in the system $\{B_i\}$; and the slave linkage rod length error $\Delta l_i$ and the dynamic platform hinge point position error $\Delta c_{io}$. These sources of error can be fully compensated using kinematic calibrations. Here, the dynamic platform hinge point position error $\Delta c_{io}$ has been considered a compensable error, so the dynamic platform hinge point position error $\Delta c_{io}$ in the non-compensable error is ignored; the error model equation in which $\varepsilon_c$ is the non-compensable error with 21 terms. Non-compensable error source are the component $\Delta a_{ioy}$ of the position error $\Delta a_{io}$ of the hinge point of the fixed platform on the $y_i$ axis of the system $\{O\}$; the components $\theta_{A_{ix}}$, $\theta_{A_{iz}}$ of the attitude error vector $\theta_{A_i}$ on the $x_i$ and $z_i$ axes of the system $\{O_i\}$; the component $\Delta b_{iy}$ of the revolute position error vector $\Delta b_i$ on the $y_{B_i}$ axis in the system $\{B_i\}$; the components $\theta_{B_{ix}}$ and $\theta_{B_{iz}}$ of the attitude error vector $\theta_{B_i}$ on the $x_{B_i}$ and $z_{B_i}$ axes in the system $\{A_i\}$; the component $\theta_{B'_{ix}}$ of the attitude error vector $\theta_{B'_i}$ on $x_{B_i}$ in the tether $\{B_i\}$. Non-compensable errors need to be strictly controlled during the manufacturing and assembly process.

## 5. Prototype Modeling and Error Analysis

### 5.1. Build ADAMS 3D Model

Using ADAMS for the kinematic simulation of hybrid robots, through simulation, the tool position and parameters of each motion sub can be obtained during the robot's end motion. In Figure 1, a 3D model is created in SOLIDWORKS 2020. Since the model imported from SOLIDWORKS into ADAMS 2016 cannot be modified directly, the error parameters cannot be added to the model. Therefore, it is necessary to create a 3D simplified parametric model of the parallel mechanism of the hybrid robot in ADAMS that parameterizes all the

key position coordinates that can determine the structure of the parallel mechanism. The geometric structure parameters of the parallel mechanism are shown in Table 1, and the parameterized model is shown in Figure 5. In this paper, the error model established is an error mapping model for static errors, and the elastic deformation of the mechanism and other non-geometric error factors are not considered. Therefore, the parameterized model of the parallel mechanism established in ADAMS is rigid.

**Table 1.** Parallel mechanism structure parameters.

|  | Fixed Platform Radius | Dynamic Platform Radius | Slave Linkage Length | Guide Rail Length |
|---|---|---|---|---|
| Size (mm) | 250 | 135 | 570 | 500 |

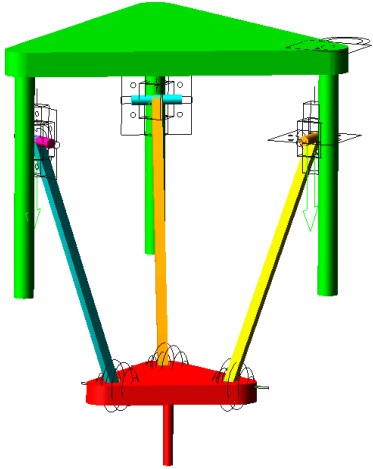

**Figure 5.** Parametric model of the parallel mechanism.

In ADAMS, the end of the robot is driven by point drive. The spatial curve
$$\begin{cases} x = 30\cos(\pi t) \\ y = 30\sin(\pi t) \\ z = 30t \end{cases}$$
is selected as the motion trajectory of the robot's end, measuring the displacement-time data of each drive sub. The measured data is generated into a spline curve, as shown in Figure 6. Then the spline function (AKISPL) is used to drive the motion of the parallel mechanism. The measured trajectory of the end of the robot is shown in Figure 7. By overlapping the measured motion trajectories of two different drive modes, the correctness of the established parameterized model is verified.

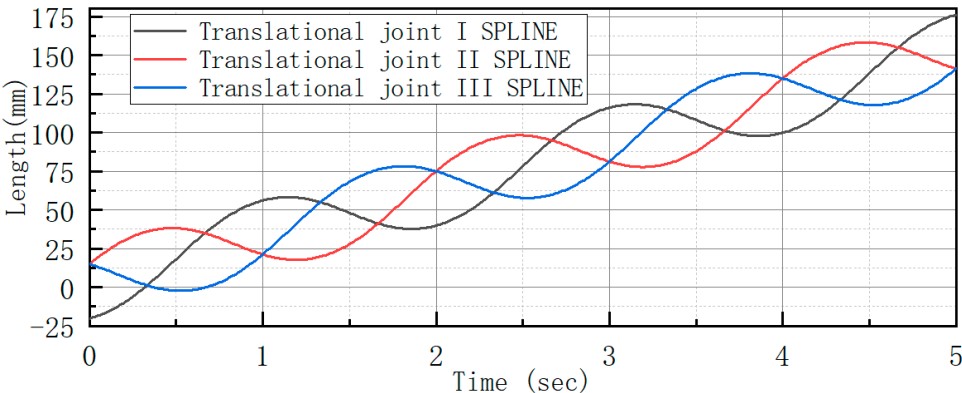

**Figure 6.** Spline curves for three drive subs.

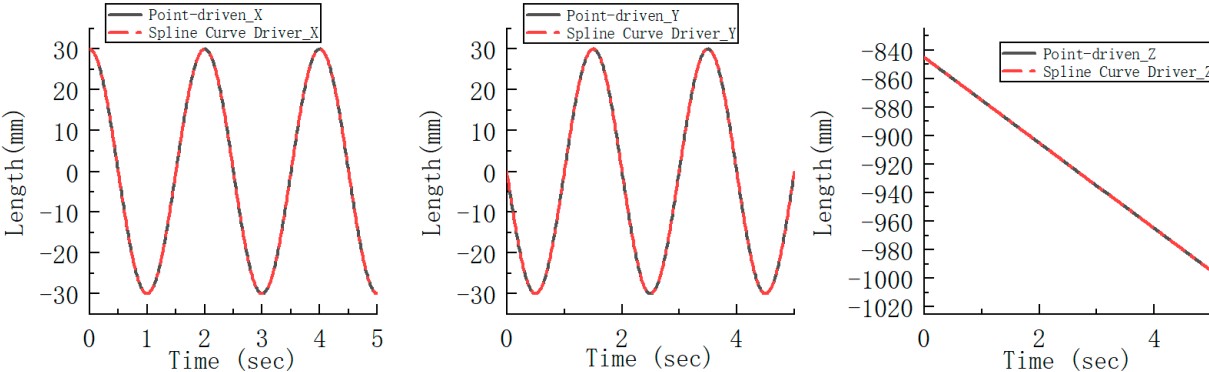

**Figure 7.** End-motion trajectory curves of two drive types.

On the established parametric model, when the simulation is performed again to measure the displacement at the end of the robot, it can truly reflect the motion transmitted to the end by the joint action of each branch chain.

### 5.2. Posture Error Analysis

#### 5.2.1. Effect of a Single Geometric Error Source on Robot End Position Error

Based on the geometric error sources analyzed above and the established error mapping model, error analysis is performed on the separated compensable error sources. Identify the compensable error sources that have the most significant effect on the robot's end posture error. Here, we will investigate how a hybrid robot parallel mechanism's hinge point position error of the fixed and dynamic platforms, the translational joint initial position error, and the revolute joint gap error affect the robot end of the robot.

Assuming that the error of each geometric structure parameter of the mechanism is 0.1 mm, the key positions of the parallel mechanism are parameterized by using the error parameters. These key positions are the hinge point position of the dynamic fixed platform, the initial linkage length of the driven linkage, the translational joint initial position, and the revolute joint gap. Then, the kinematic simulation of the robot is performed to measure the displacement trajectory of the robot's end. Compare the motion trajectories containing errors with the original motion trajectories not containing errors, and finally derive the effect of each error source on the robot's end motion.

When only dynamic and fixed platform hinge position error ($\Delta c_{iO}$, $\Delta a_{iO}$) is considered, the robot end motion trajectory error curve is shown in Figure 8. (FS_X represents the error curve of the hinge point position error of the dynamic fixed platform on the robot end in the X-direction):

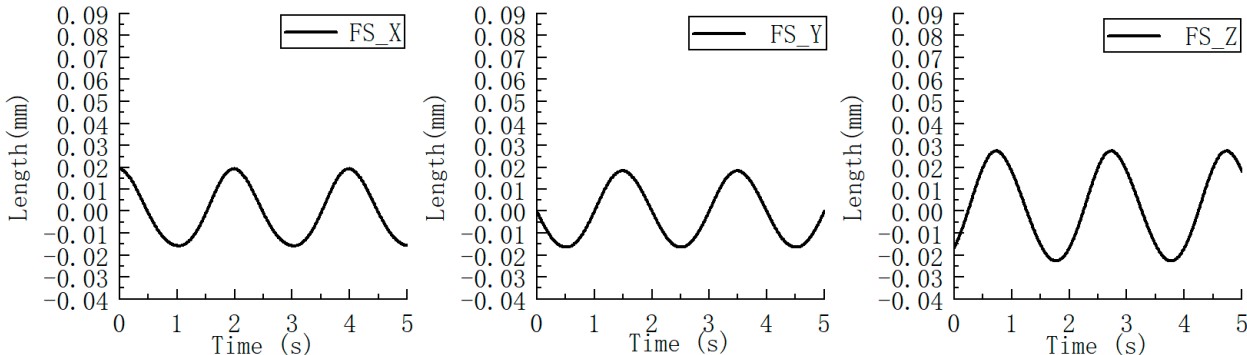

**Figure 8.** Dynamic and fixed platform hinge position error curve.

When only the translational joint initial position error ($\Delta q_i$) is considered, the robot end motion trajectory error curve is shown in Figure 9 (T_X represents the error curve of the translational joint initial position error on the robot end in the X-direction):

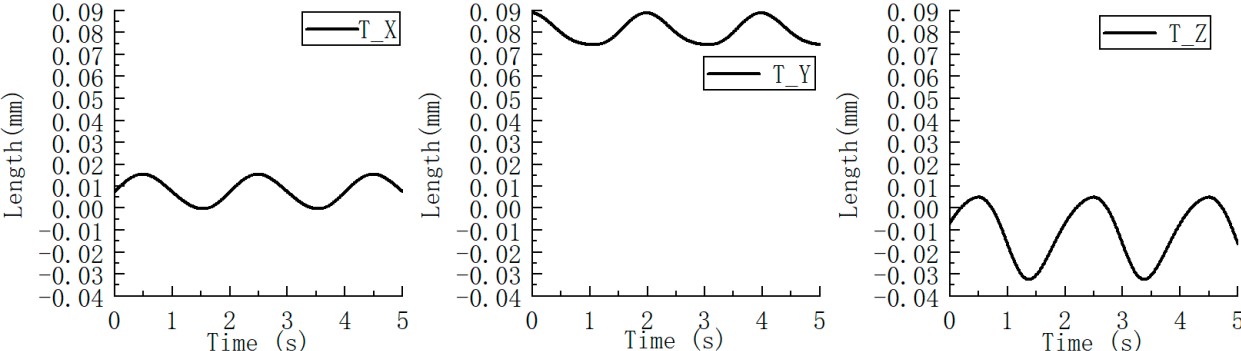

**Figure 9.** Translational joint initial position error curve.

When only the revolute joint gap error ($\Delta b_i$) is considered, the robot end motion trajectory error curve is shown in Figure 10 (R_X represents the effect of the revolute joint gap error on the X-direction of the robot end):

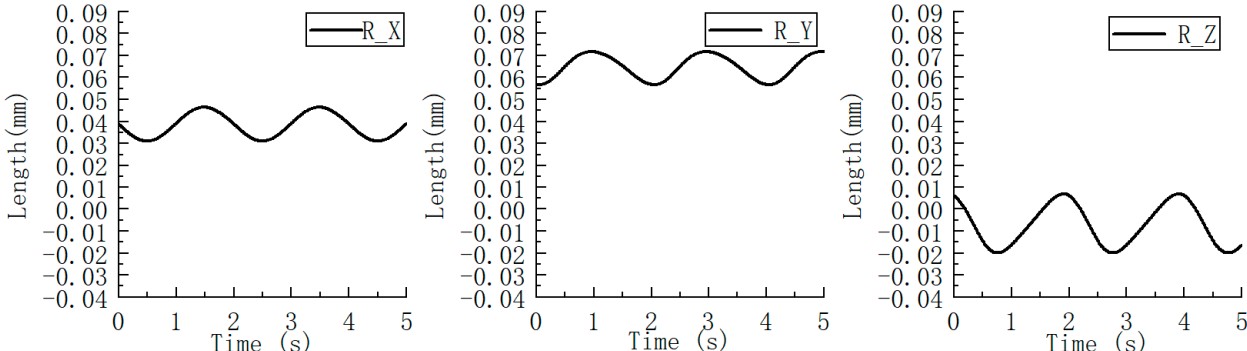

**Figure 10.** Revolute joint gap error curve.

When only the rod length error ($\Delta l_i$) is considered, the robot end motion trajectory error curve is shown in Figure 11 (L_X represents the effect of the rod length error on the X direction of the robot end):

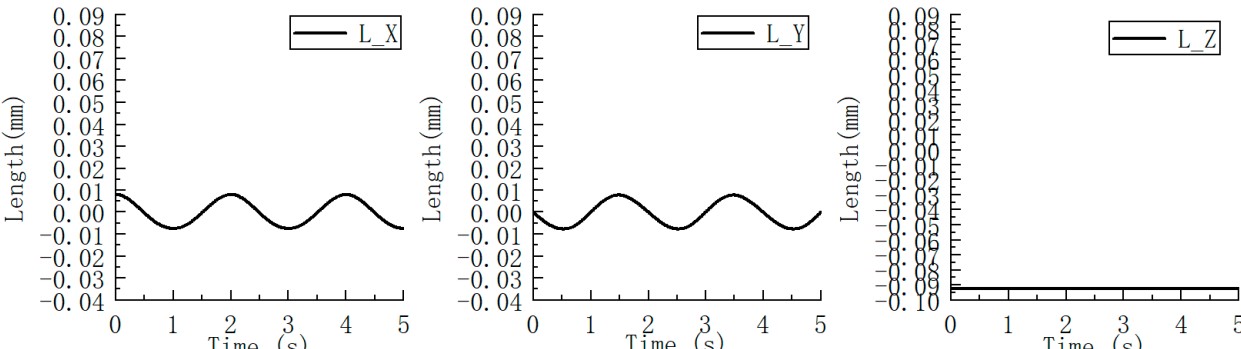

**Figure 11.** Rod length error curve.

Above is the error profile of each error simulated by ADAMS. As can be seen from Figure 8, the maximum error of the dynamic and fixed platform hinge position on the end of the robot is 0.0352 mm in the X direction of motion, 0.035 mm in the Y direction, and 0.0501 mm in the Z direction. As can be seen from Figure 9, the maximum error of the initial position error of the moving sub on the end of the robot is 0.0157 mm in the X direction of motion, 0.0144 mm in the Y direction, and 0.0373 mm in the Z direction. As can be seen from Figure 10, the maximum error of the revolute joint gap error on the end of the robot is 0.0154 mm in the X direction of motion, 0.015 mm in the Y direction, and 0.0268 mm in the Z direction. As can be seen from Figure 11, the maximum error of the rod length error on the end of the robot is 0.0155 mm in the X direction of motion, 0.0154 mm in the Y direction, and 0.0003 mm in the Z direction. As a result, it can be concluded that the dynamic and fixed platform hinge position error has the most significant effect on the robot end position, and the rod length error has the least effect on the Z direction of the end motion. From the four sets of plots, it can be seen that when the robot's end trajectory changes regularly, the effect of each error source on the end position also changes into a regular pattern.

To verify the influence of each error source on the position of the robot end, the simulation was verified several more times in the error (0.1 mm–0.5 mm) range. When the error is 0.3 mm, the maximum error of the dynamic and fixed platform hinge position on the end of the robot in X, Y, and Z directions are 0.1057 mm, 0.1049 mm, and 0.1504 mm, respectively; the maximum errors of the initial position error of the moving sub to the end of the robot in the X, Y, and Z directions are 0.0473 mm, 0.0433 mm, and 0.1118 mm, respectively; the maximum error of rotating sub gap error on the end of the robot in X, Y, and Z directions is 0.0461 mm, 0.045 mm, and 0.084 mm respectively; the maximum error of the rod length on the end of the robot in X, Y, and Z directions is 0.0525 mm, 0.0526 mm, and 0.00032 mm, respectively. When the error is 0.5 mm, the maximum error of the dynamic and fixed platform hinge position on the end of the robot in the X, Y, and Z directions are 0.1761 mm, 0.1748 mm, and 0.2508 mm, respectively; the maximum errors in the X, Y, and Z directions of the initial position error of the moving sub for the end of the robot are 0.0789 mm, 0.0722 mm, and 0.1862 mm, respectively; the maximum errors of the rotating sub gap errors on the X, Y, and Z directions at the end of the robot are 0.0768 mm, 0.075 mm, and 0.1341 mm, respectively; the maximum error of the rod length on the end of the robot in X, Y, and Z directions is 0.0876 mm, 0.0878 mm, and 0.0005 mm, respectively. From this, it can be concluded that the dynamic and fixed platform hinge position error has the most significant effect on the end position of the robot; the rod length error is second only to the dynamic and fixed platform hinge position error in the X and Y directions of the end motion, but has the least effect in the Z direction; and the effect of the initial position error of the translational joint and the revolute joint gap error on the end position is basically equivalent.

The equation of the motion trajectory of the end of the parallel mechanism part $\begin{cases} x = 15t\cos(\pi t) \\ y = 15t\sin(\pi t) \\ Z = 15t \end{cases}$ is set in space again. The simulation is performed again using the same method steps.

Similarly, setting the error of each structural parameter to 0.1 mm, the following error curves can be obtained by simulating the robot using the same methodological steps.

When only moving and fixed platform hinge position error ($\Delta c_{iO}$, $\Delta a_{iO}$) is considered, the robot end motion trajectory error curve is shown in Figure 12:

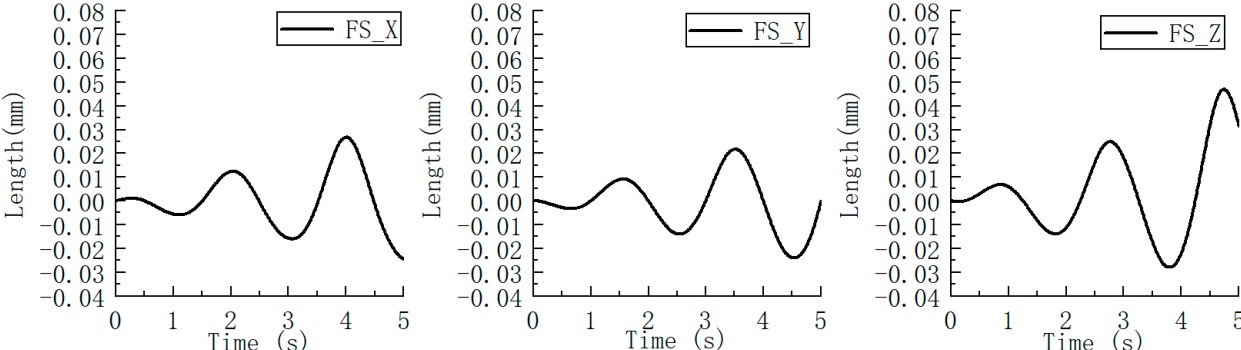

**Figure 12.** Dynamic and fixed platform hinge position error curve. (Second trajectory).

When only the initial position error ($\Delta q_i$) of the translation is considered, the robot end motion trajectory error curve is shown in Figure 13:

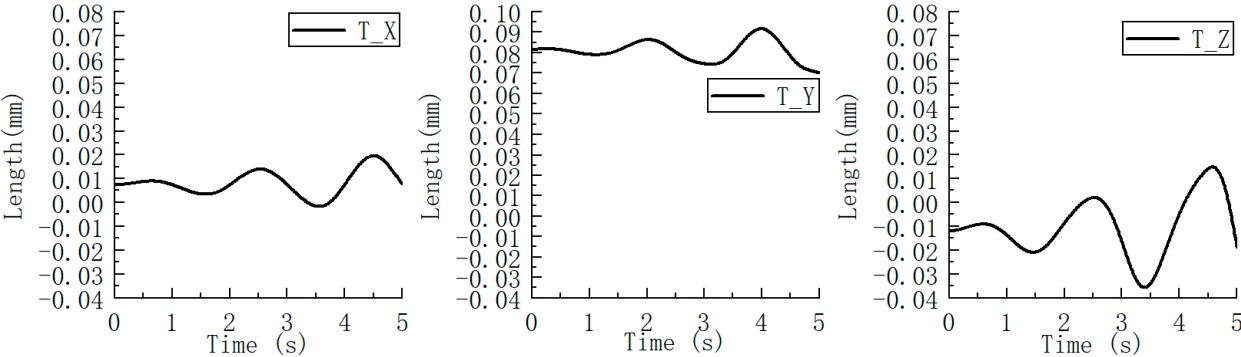

**Figure 13.** Translational joint initial position error curve. (Second trajectory).

When only the revolute joint gap error ($\Delta b_i$) is considered, the robot end motion trajectory error curve is shown in Figure 14:

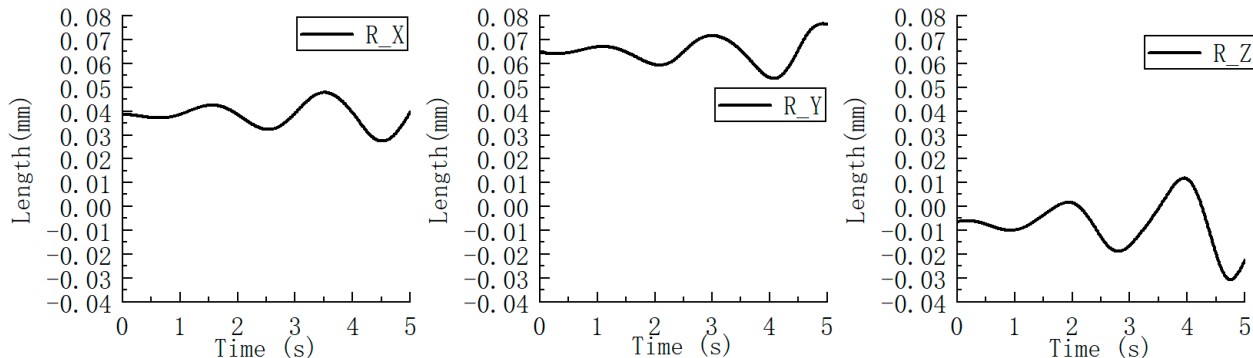

**Figure 14.** Revolute joint gap error curve. (Second trajectory).

When only the rod length error ($\Delta l_i$) is considered, the robot end motion trajectory error curve is shown in Figure 15:

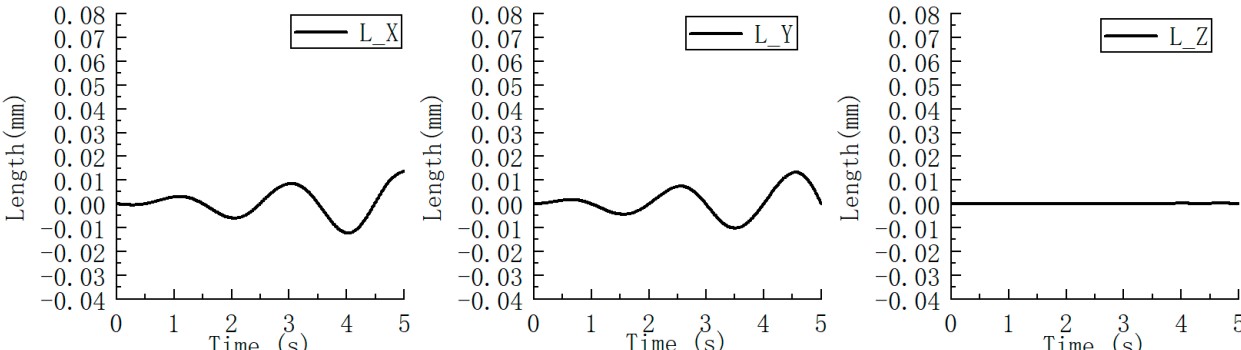

**Figure 15.** Rod length error curve. (Second trajectory).

Based on the error curve figures derived from the above simulation, the same conclusion can be drawn for the two spatial motion trajectories. The dynamic and fixed platform hinge position error has the most significant effect on the end position of the robot, and the rod length error has the least effect in the Z direction of the end motion. The effect of the translational joint initial position error is smaller than the rod length error, and the revolute joint gap error is smaller than the effect of the translational joint initial position error, but the effects of the two error sources are basically equal. The error curve simulated by the second trajectory equation shows that the magnitude of the error curve is gradually increasing, so it can also be concluded that the effect of each error source on the end position of the robot will increase with the increase of the end trajectory.

### 5.2.2. Effect of Coupling Multiple Geometric Error Sources on Robot End Position Error

The geometric errors of the parallel robot are coupled with each other to form the spatial error of the whole machine. To analyze the effect of the coupling between the geometric error terms on the robot's end position based on the conclusions drawn in Section 5.2.1, this summary investigates the coupling of the error sources. Separately, all error sources, two error sources, and three error sources are coupled, and the calculated results are compared with the results of coupling all error sources. The results of the calculations are shown in Figure 16, Figure 17 and Figure 18, respectively.

The error profiles after coupling all error sources are plotted in Figure 16.

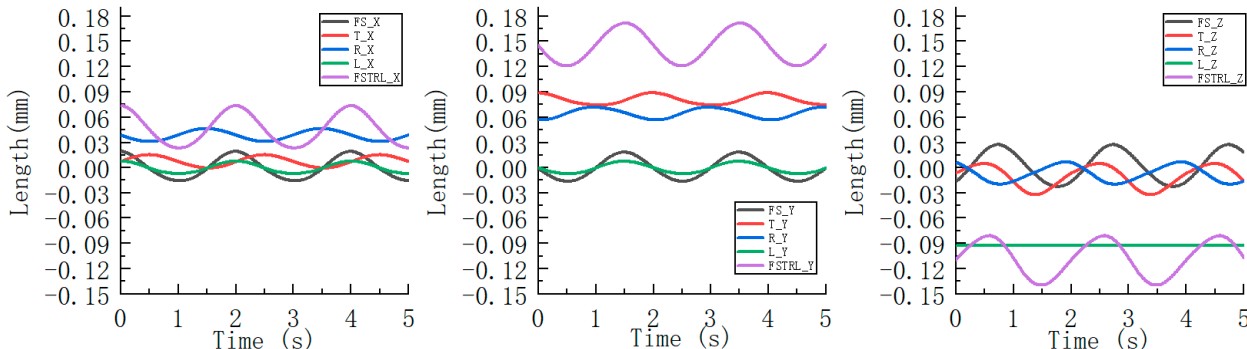

**Figure 16.** Effect of coupling all geometric error sources on robot end position error.

As can be seen from Figure 16, for the X and Y coordinates of the robot end, the position coordinate errors of the robot end caused by the translational joint initial position error and the revolute joint gap error are in opposite directions, so the two can basically cancel each other out. The position coordinate errors of the robot end caused by the remaining two errors (the dynamic and fixed platform hinge position error and the rod length error of the slave linkage) are basically in the same direction, so the comprehensive position error increases after superposition. As for the Z coordinate of the robot end, the influence of

the translational joint initial position error on the Z coordinate of the robot end is greater than that of the revolute joint gap error, so the two cannot cancel each other. Among them, the direction of the position coordinate error of the robot end caused by the translational joint initial position error, the dynamic and fixed platform hinge position error, and the rod length error are basically the same. After superposition, the comprehensive position error increases. In the figure, it can be seen that the coupling of several error sources is not a simple superposition, and the error offset phenomenon can occur during the coupling of error sources. (In the legend, FS represents the dynamic and fixed platform hinge position error, T represents the translational joint initial position error, R represents the revolute joint gap error, L represents the slave linkage rod length error, and FSTRL represents the error after coupling the four error sources, and the same representation is applied in the subsequent legends.)

The error profile after coupling the two error sources is plotted in Figure 17.

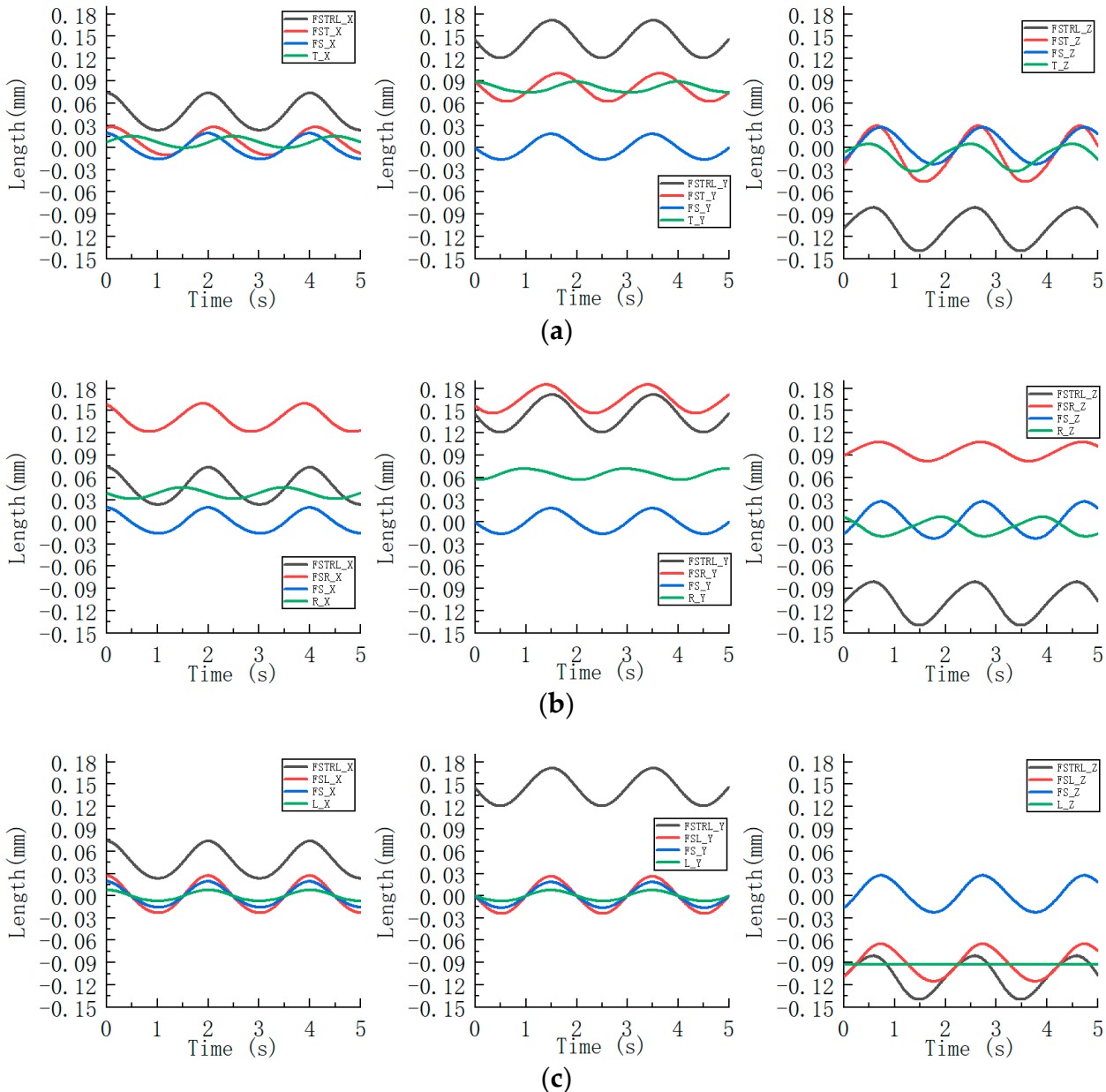

**Figure 17.** *Cont.*

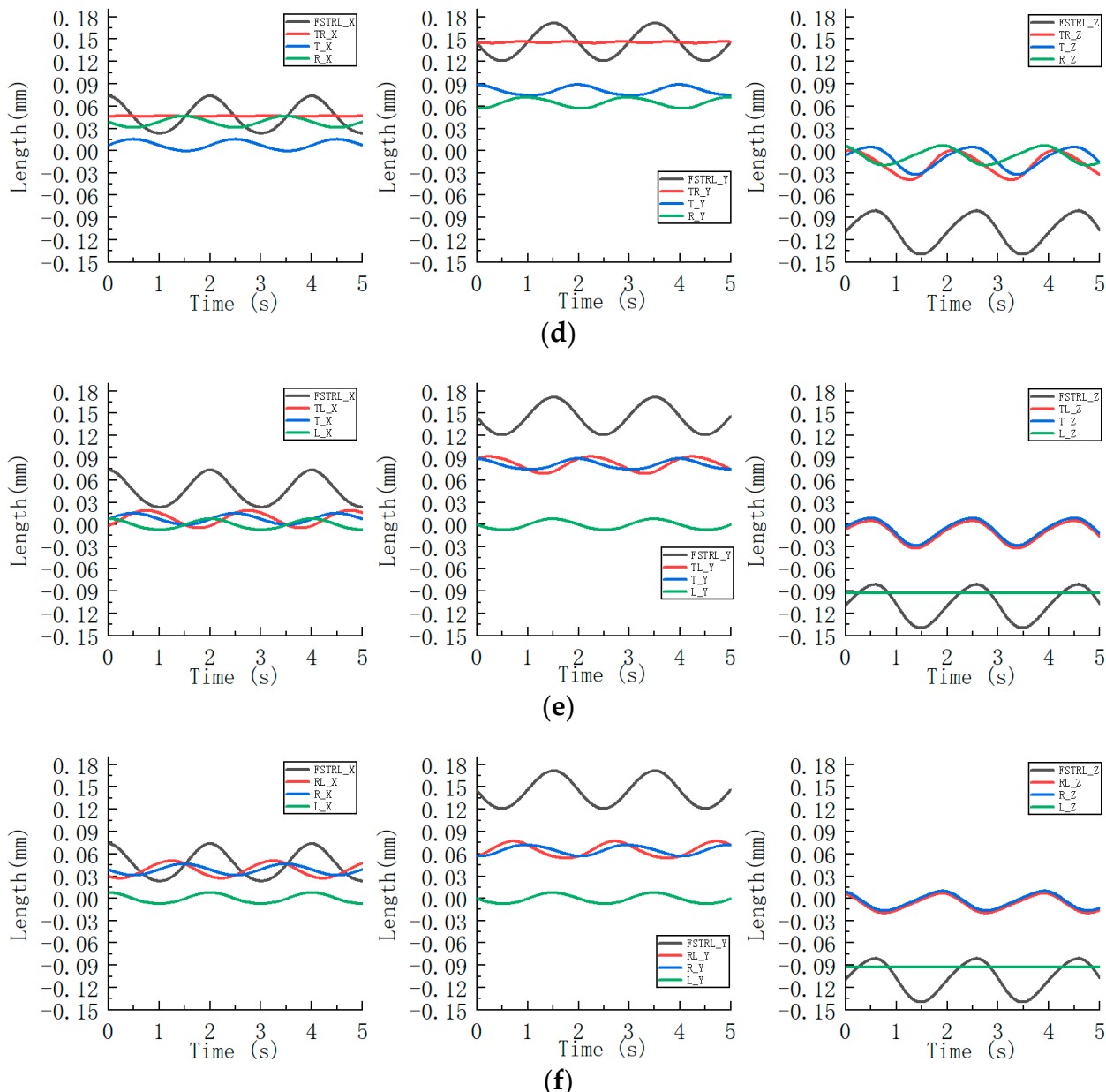

**Figure 17.** Effect of coupling two geometric error sources on robot end position error. (**a**) $(\Delta c_{iO}, \Delta a_{iO})$ coupled with $(\Delta q_i)$; (**b**) $(\Delta c_{iO}, \Delta a_{iO})$ coupled with $(\Delta b_i)$; (**c**) $(\Delta c_{iO}, \Delta a_{iO})$ coupled with $(\Delta l_i)$; (**d**) $(\Delta q_i)$ coupled with $(\Delta b_i)$; (**e**) $(\Delta q_i)$ coupled with $(\Delta l_i)$; (**f**) $(\Delta b_i)$ coupled with $(\Delta l_i)$.

In Figure 17, from Figure 17a, it can be seen that after the coupling of the dynamic and fixed platform hinge point position error $(\Delta c_{iO}, \Delta a_{iO})$ and the translational joint initial position error $(\Delta q_i)$, for the X, Y, and Z coordinates of the robot end, the change patterns of the posture error curves at the robot end caused by the two errors are basically the same, and the directions of the posture errors at the robot end caused by them are also basically the same most of the time. After the superposition, the integrated posture error at the robot end increases. The dynamic and fixed platform hinge point position error dominates. These two error sources coupled in the Z coordinate have a larger fluctuation in the integrated position error curve than the fluctuation in the integrated position error curve of all the error sources coupled.

In Figure 17, from Figure 17b, it can be seen that after the coupling of the dynamic and fixed platform hinge point position error $(\Delta c_{iO}, \Delta a_{iO})$ and the revolute joint gap error $(\Delta b_i)$. For the X-coordinate and Y-coordinate of the robot end, the change rule of the posture error

curve of the robot end caused by the two errors is basically the same, and the direction is also basically the same for most of the period. So, after the superposition, the integrated posture error of the robot end is increased. The dynamic and fixed platform hinge point position error dominates. For the Z coordinate at the end of the robot, the two errors cause the positional errors at the end of the robot to be in opposite directions, and the integrated positional errors are reduced after superposition. The integrated positional errors in X, Y, and Z directions are smaller than the integrated positional errors after coupling all error sources.

In Figure 17, from Figure 17c, it can be seen that after the coupling of the dynamic and fixed platform hinge point position error ($\Delta c_{iO}$, $\Delta a_{iO}$) and the slave linkage rod length error ($\Delta l_i$). For the X and Y coordinates of the robot end, the two errors cause the same direction of the posture error of the robot end, and the integrated posture error increases after the superposition. Since the error of the slave linkage length is very small in the Z coordinate, the changing pattern of the integrated posture error after superposition is similar to the posture error curve of the dynamic and fixed platform hinge point position errors. For the three coordinates, the influence of the dynamic and fixed platform hinge point position errors is dominant.

In Figure 17, from Figure 17d, it can be seen that after the translational joint initial position error ($\Delta q_i$) and the revolute joint gap error ($\Delta b_i$) are coupled, for the X and Y coordinates of the robot end, the two errors cause the position error of the robot end to be in opposite directions. After superposition, the integrated position error is significantly reduced and tends to a horizontal line. For the Z coordinate at the end of the robot, the error increases after coupling because the effects of the two errors are not cancelable. The integrated positional errors in X, Y, and Z directions are smaller than the integrated positional errors after coupling all error sources.

In Figure 17, from Figure 17e, it can be seen that after the translational joint initial position error ($\Delta q_i$) is coupled with the slave linkage rod length error ($\Delta l_i$), for the X and Y coordinates at the end of the robot, the integrated positional error does not change significantly after the two errors are coupled. For the Z coordinate of the robot end, the error curve tends to be in a horizontal line because of the small influence of the rod length error in the Z direction. Therefore, the change rule of the integrated position error curve after the coupling of the two errors is similar to the change rule of the error curve caused by the translational joint initial position error and tends to coincide. For the three coordinates, the influence of the translational joint initial position error dominates.

In Figure 17, from Figure 17f, it can be seen that after coupling the revolute joint gap error ($\Delta b_i$) with the slave linkage rod length error ($\Delta l_i$), for the X and Y coordinates at the end of the robot, the integrated posture error does not change significantly after the two errors are coupled. For the Z coordinate at the end of the robot, the rod length error has little effect in the Z direction and tends to be a horizontal line. Therefore, the change rule of the integrated position error curve after the coupling of the two errors is similar to the change rule of the error curve caused by the revolute joint gap error and tends to overlap. The integrated posture error after coupling the translational joint initial position error and rod length error and the integrated posture error after coupling the revolute joint gap error and rod length error are basically comparable in the X and Y coordinates at the end of the robot, but in the Z coordinate direction, the integrated posture error of the former has a more significant effect on the end of the robot.

The error profile after coupling the three error sources is plotted in Figure 18.

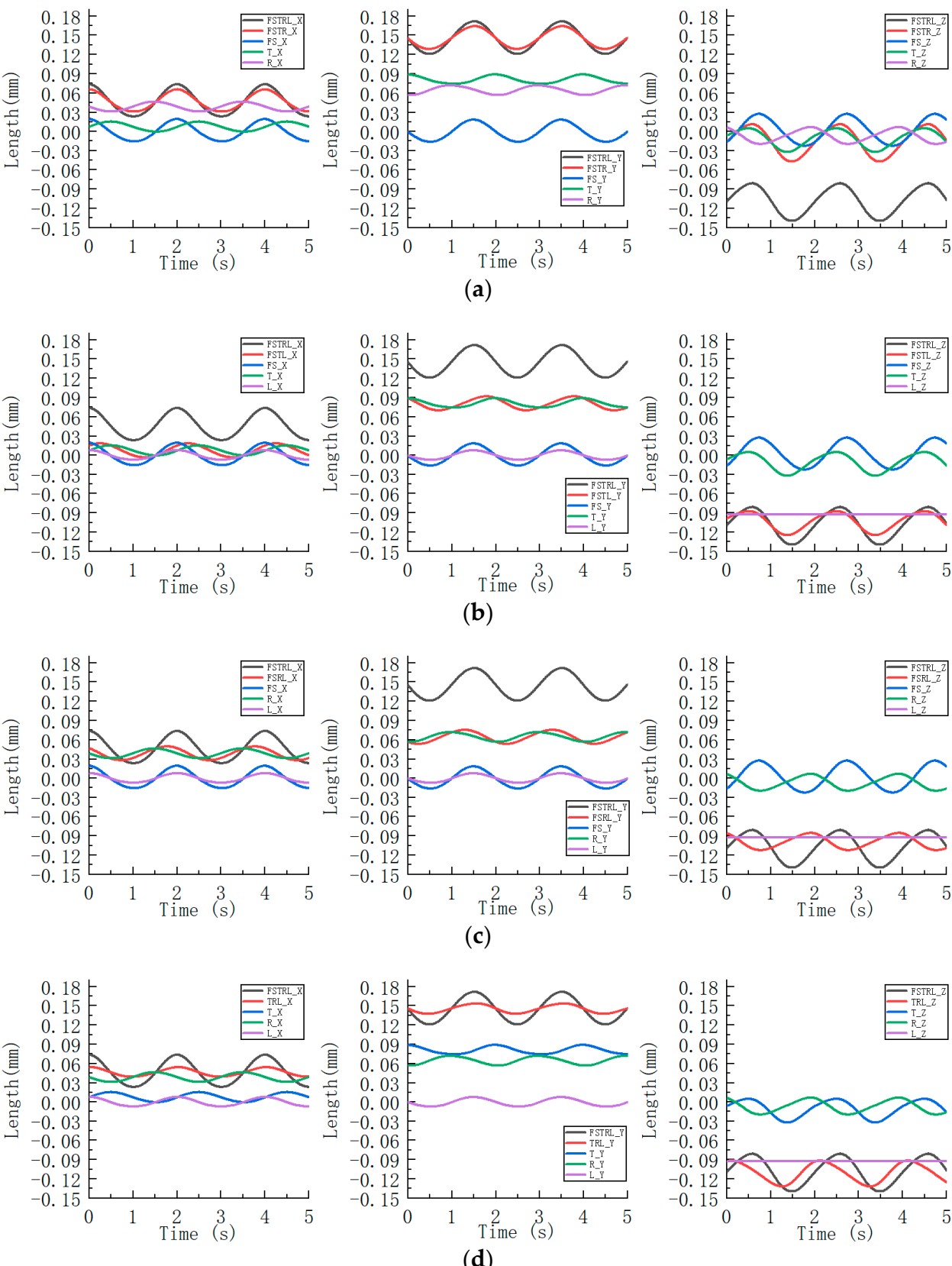

**Figure 18.** Effect of coupling three geometric error sources on robot end position error. (**a**) ($\Delta c_{iO}$, $\Delta a_{iO}$), ($\Delta q_i$) and ($\Delta b_i$) coupled; (**b**) ($\Delta c_{iO}$, $\Delta a_{iO}$), ($\Delta q_i$) and ($\Delta l_i$) coupled; (**c**) ($\Delta c_{iO}$, $\Delta a_{iO}$), ($\Delta b_i$) and ($\Delta l_i$) coupled; (**d**) ($\Delta q_i$), ($\Delta b_i$) and ($\Delta l_i$) coupled.

In Figure 18, from Figure 18a, it can be seen that after the coupling of the dynamic and fixed platform hinge position error ($\Delta c_{iO}$, $\Delta a_{iO}$), the translational joint initial position error ($\Delta q_i$) and the revolute joint gap error ($\Delta b_i$), for the X and Y coordinates of the end of the robot, the error curves caused by the translational joint initial position error and the revolute joint gap error are in opposite directions, so the effects can be canceled out. So, the changing pattern of the integrated posture error curve after the coupling of the three errors is similar to that of the dynamic and fixed platform hinge position errors. For the Z coordinate of the robot end, the integrated position error increases after superposition because the direction of the position error curve of the robot end is caused by the dynamic and fixed platform hinge position error, and the translational joint initial position error is basically the same.

In Figure 18, from Figure 18b, it can be seen that after the coupling of the dynamic and fixed platform hinge position error ($\Delta c_{iO}$, $\Delta a_{iO}$), the translational joint initial position error ($\Delta q_i$) and the rod length error ($\Delta l_i$) of the slave linkage, for the X and Y coordinates of the robot end, the direction of the posture error curve of the robot end caused by the dynamic and fixed platform hinge position error and the rod length error is basically the same, but the translational joint initial position error is opposite to the other two errors. Therefore, the integrated position error after coupling is smaller than the position error of the robot end caused by the dynamic and fixed platform hinge position errors. For the Z coordinate at the end of the robot, the integrated posture error after the coupling of the three error sources is smaller than the posture error at the end of the robot caused by the dynamic and fixed platform hinge position errors. The integrated positional error in X, Y, and Z directions is smaller than the integrated posture error after coupling all error sources.

In Figure 18, from Figure 18c, it can be seen that after the coupling of the dynamic platform hinge position error ($\Delta c_{iO}$, $\Delta a_{iO}$), revolute joint gap error ($\Delta b_i$), and slave rod length error ($\Delta l_i$), for the X and Y coordinates of the end of the robot, the changing pattern and direction of the posture error curve at the end of the robot caused by the dynamic platform hinge position error and the slave linkage rod length error are basically the same, but the revolute joint gap error is in the opposite direction of the error curve caused by the former two error sources. Therefore, the integrated posture error after superposition is smaller than the effect of the single error source of the dynamic platform hinge position error. For the Z coordinate at the end of the robot, the integrated positional error is reduced after superposition because the rod length error has little effect, and the direction of the position error curve at the end of the robot is caused by the revolute joint gap error, while the dynamic and fixed platform hinge position errors are opposite. In the three directions of the robot end, the integrated posture error after coupling these three error sources is significantly smaller than the integrated posture error after coupling all error sources.

In Figure 18, from Figure 18d, it can be seen that after the translational joint initial position error ($\Delta q_i$), the revolute joint gap error ($\Delta b_i$) and the slave linkage rod length error ($\Delta l_i$) are coupled. For the X and Y coordinates of the robot end, in the opposite direction of the position error curve of the robot end caused by the translational joint initial position and revolute joint gap error. So, the integrated posture error of the three errors after superposition is reduced. For the Z coordinate at the end of the robot, the rod length error has little effect, and the position error curve at the end of the robot is caused by the translational joint initial position error, and the rotating sub-gap error is basically in the same direction for most of the time. Therefore, the integrated position error increases after the superposition of the three errors.

## 6. Discussion

The experimental results show that, among the individual error sources, the dynamic and fixed platform hinge point position error has the most significant effect on the robot end posture, the slave linkage rod length error has the least effect on the robot end error in the Z coordinate direction, and translational joint initial position error and revolute joint gap error have the same effect on the robot end posture. The simulation of different spatial

trajectories also shows that the influence of each error source on the end posture of the robot gradually increases with the increase in the end trajectory.

Among the integrated positional errors are multiple error sources coupled to the posture error at the end of the robot. When two error sources are coupled, the dynamic and fixed platform hinge position error is dominant in the integrated posture error, which is coupled with the translational joint initial position error, the revolute joint gap error, and the slave linkage rod length error, respectively. The integrated posture error after the coupling of the translational joint initial position error and the revolute joint gap error is very small in the X and Y coordinate directions, but the error increases in the Z coordinate. In the integrated posture error of the slave linkage rod length error coupled with the translational joint initial position error and the rotating sub clearance error, respectively, the changes in X and Y coordinates are not obvious, but the integrated posture error in the Z coordinate direction after its coupling with the translational joint initial position error is larger than that after coupling with the revolute joint gap error. In the coupling of the three error sources, when there is an integrated positional error of the dynamic and fixed platform hinge point position errors, the dynamic and fixed platform hinge point position errors dominate, and the errors in the Z coordinate are larger than those in the X and Y coordinates. When the integrated positional error is without the dynamic and fixed platform hinge point position, the errors in the X and Y coordinate directions do not vary obviously, and the translational joint initial position error dominates the integrated positional error in the Z coordinate direction. Separate studies of single error sources and the coupling of multiple error sources yielded more rigorous results.

Therefore, when designing this hybrid robot, attention should be paid to the dynamic and fixed platform hinge point positions and translational joint initial positions. The machining and manufacturing accuracy of these two error sources and the assembly accuracy should be strictly controlled so that the designed robot can complete the basic surface machining requirements.

However, in practical applications, there are still some errors in the manufactured parts. These errors can add up and ultimately cause the actual position of the robot end to deviate from the theoretical position, reducing the machining accuracy of the robot. Therefore, in future research, the calibration of this hybrid robot will continue to be investigated in the direction of further improving the machining accuracy of the robot.

## 7. Conclusions

This paper presents an error analysis of the designed hybrid robot. Through the error analysis, the key error factors affecting the end position error of the mechanism can be clarified, which can guide the manufacturing and assembly of hybrid robots. Firstly, the error sources of the robot are traced, and the formation principle of each error source and the number of error sources are defined. Based on the analyzed error sources, the error mapping model is established for the parallel mechanism part of the hybrid robot by using the closed-loop vector method and the first-order perturbation method. The compensable and non-compensable error sources affecting the end posture error of the hybrid robot are separated based on the mapping property of the 6th-order velocity Jacobi matrix. Finally, we conducted simulation experiments for error analysis of the error sources. Investigated the effect of a single error source and the coupling of multiple error sources on the robot's end position error. The error analysis identifies the most significant factors affecting the robot's end posture error. When designing this hybrid robot, focus on the error sources that significantly affect the robot's end so that the robot can complete the basic surface processing requirements.

**Author Contributions:** Conceptualization, H.S.; methodology, L.D. and H.S.; software, L.D.; validation, L.D., H.Z. and X.W.; formal analysis, L.D.; investigation, H.S. and L.D.; resources, H.S.; data curation, L.D., H.Z. and X.W.; writing—original draft preparation, L.D.; writing—review and editing, H.S.; visualization, L.D., H.Z. and X.W.; supervision, H.S.; project administration, H.S.; funding acquisition, H.S. All authors have read and agreed to the published version of the manuscript.

**Funding:** Major Project of Yunnan Provincial Science and Technology Department: Research on Key Technology of Industrial Robot and its Application Demonstration Project in Intelligent Manufacturing: 202002AC080001.

**Data Availability Statement:** Data are unavailable due to privacy or ethical restrictions.

**Conflicts of Interest:** The authors declare no conflict of interest.

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
