# Peer review of "Error Analysis of a New Five-Degree-of-Freedom Hybrid Robot"

_actuators, doi:10.3390/act12080324_

Round 1

Reviewer 1 Report (Previous Reviewer 2)

This revised version of the paper is much improved. Therefore, I recommend the publication of the paper.

Author Response

We sincerely appreciate the valuable comments. We have checked the literature carefully and added more recent references to the Introduction part of the revised manuscript.

Reviewer 2 Report (Previous Reviewer 1)

In this paper, the authors presented a new five-degree-of-freedom hybrid robot for complex surface machining and an error analysis study for this robot.

The authors have improved their manuscript, according to reviewers comments. Unfortunately, some of sentences are too long and difficult to be understood. So, before accepting their manuscript for publication, the authors should careful read it and to rearange the sentences (shorter and more clear sentences are recommended).

We give two examples:

„In this paper, we design a new five-degree-of-freedom hybrid robot for complex surface machining, whose main body (3PRS parallel mechanism) is the key component responsible for complex surface machining.”

„There are many error factors affecting the end pose accuracy of the robot, and the error sources can be classified into different types according to different causes of formation, and the error sources can be classified into static and dynamic errors according to the time characteristics [10].”

English could be improved.

Author Response

Author's Response: We sincerely appreciate the valuable comments. We have tried our best to revise the English version of the article. The article was read line by line, and the long sentences in the article were modified to be presented in shorter, clearer sentences.

We sincerely thank the reviewers once again for your comments.

Reviewer 3 Report (New Reviewer)

Comments and Suggestions for Authors

The manuscript deals with a interesting subject.

The language and presentation should be improved.  The authors repeatedly use very long sentences, which makes it difficult for the reader to understand the information provided by the authors.

In the article, the authors presented an error analysis of the designed hybrid robot. Unfortunately, the analysis applies only to the robot model, and the authors did not refer to the actual device or components in any way.

The analyses conducted indicate the construction factors affecting manipulator end position errors and can serve as a guideline for robot designers. The simulations carried out make it possible to study the influence of single error sources and the coupling of multiple error sources on the manipulator end position error.

The analysis carried out and the conclusions in the article are correct. It should be added here that in manipulators (in most cases) the errors that occur in individual components add up. Hence, it is difficult to achieve high accuracy in these devices. Given the intention to use the designed device in machining applications, there is a high probability that the manipulator will perform complex movements. In particular that the manipulator under consideration has a parallel design.

Figures 6 and 7 should be improved because they are unclear.

Author Response

1.The language and presentation should be improved.  The authors repeatedly use very long sentences, which makes it difficult for the reader to understand the information provided by the authors.

Author's Response: We sincerely appreciate the valuable comments. We have tried our best to revise the English version of the article. The article was read line by line, and the long sentences in the article were modified to be presented in shorter, clearer sentences.

2.In the article, the authors presented an error analysis of the designed hybrid robot. Unfortunately, the analysis applies only to the robot model, and the authors did not refer to the actual device or components in any way.

Author's Response: We sincerely appreciate the valuable comments. Since our robot is still in the design stage, the physical robot has not yet been manufactured. The purpose of this paper is to analyze the error of the robot at the robot design stage, to find out the factors that have a significant impact on the end of the robot, so that the error can be minimized as much as possible at the design stage and the critical parts can be strictly machined and manufactured.

3.The analyses conducted indicate the construction factors affecting manipulator end position errors and can serve as a guideline for robot designers. The simulations carried out make it possible to study the influence of single error sources and the coupling of multiple error sources on the manipulator end position error.

Author's Response: Thank you very much for your comments on this article. The simulation experiments carried out in this paper are valuable for studying the effects of single error sources and the combined position error after coupling of multiple error sources on the robot end. Error sources that significantly affect the robot end can be identified.

4.The analysis carried out and the conclusions in the article are correct. It should be added here that in manipulators (in most cases) the errors that occur in individual components add up. Hence, it is difficult to achieve high accuracy in these devices. Given the intention to use the designed device in machining applications, there is a high probability that the manipulator will perform complex movements. In particular that the manipulator under consideration has a parallel design.

Author's Response: Thank you very much for your comments on this article. The reviewers' comments are correct. We have added to the reviewers' additions in the article accordingly. Specifically on page 22, line 683 in the article. In future research, we will continue along the direction of improving the accuracy of this robot.

5.Figures 6 and 7 should be improved because they are unclear.

Author's Response: We sincerely appreciate the valuable comments. In the text we have revised Figures 6 and 7, and the explanations of the two figures have been revised accordingly. Specifically on page 12, line 365 of the text.

We sincerely thank the reviewers once again for your comments.

Round 2

Reviewer 2 Report (Previous Reviewer 1)

Authors answered to the reviewers comments.

Minor editing of English language required.

This manuscript is a resubmission of an earlier submission. The following is a list of the peer review reports and author responses from that submission.

Round 1

Reviewer 1 Report

1. Figure 1 should be reconsidered. The robot structure is not clear. It is hidden by the robot base. Notations are also unclear.

2. More equations cannot be read. Please, correct the typos or the pdf file.

3. Figures 7 and 12 are useless if we cannot understand them.

Author Response

Dear Reviewer,

We sincerely appreciate your suggestions, and we have made changes based on the suggestions you have made.

  1. Figure 1 should be reconsidered. The robot structure is not clear. It is hidden by the robot base. Notations are also unclear.

Author's Response: We think this is an excellent suggestion. We have modified Figure 1, and the modified figure clearly presents the structure of the robot. Figure 1 is on page three, line 127 in the revised draft.

  1. More equations cannot be read. Please, correct the typos or the pdf file.

Author's Response: We sincerely apologize that the difference in editors may have caused you reading difficulties, and in the revised version we have edited the formulas in MathType and made them available as PDF files for reading.

  1. Figures 7 and 12 are useless if we cannot understand them.

Author's Response: We think this is an excellent suggestion. The purpose of Figure 7 is to verify the correctness of the parametric model created in ADAMS by measuring the overlap of the trajectories with two different driving methods. Figure 12 serves the same purpose, but Figure 12 appears redundant and has now been deleted. The explanation of Figure 7 is on page 12, line 365 of the revised draft.

The revised draft also modifies the results, which are revised to more clearly present the conclusions drawn in the article. We sincerely thank the reviewers for your valuable suggestions.

Reviewer 2 Report

The authors present in this paper an error analysis for a new five-degree-of-freedom hybrid robot. The paper topic is interesting however the paper needs some major improvements.

The reference study in the introduction is weak. More recent reference should be added to the paper.

In the introduction, only one paragraph presents the work of the authors. No comparison is made with other works. The authors should highlight the contribution of the paper.

Adams software is usually used for dynamic simulations. Here, only motion simulations are made. Why not only using the inverse kinematics for errors analysis?

The Inverse kinematics model is missing in the paper!

What are the requirements of the intended application in terms of accuracy?

How can the carried out work help designing the presented robot?

Author Response

Dear Reviewer,

We sincerely appreciate your suggestions, and we have made changes based on the suggestions you have made.

  1. The reference study in the introduction is weak. More recent references should be added to the paper.

Author's Response: We sincerely appreciate the valuable comments. We have checked the literature carefully and added more recent references to the Introduction part of the revised manuscript.

  1. In the introduction, only one paragraph presents the work of the authors. No comparison is made with other works. The authors should highlight the contribution of the paper.

Author's Response: We think this is an excellent suggestion. We have rewritten this part according to the Reviewer's suggestion. The revised article adds comparisons and also emphasizes the work done.

  1. Adams software is usually used for dynamic simulations. Here, only motion simulations are made. Why not only using the inverse kinematics for errors analysis?

Author's Response: Thank you very much for your comment. We thought about this section once again. Our insights are as follows: In ADAMS/VIEW, if the moving platform is required to move according to a certain motion law when the length of each telescopic rod changes, a point motion excitation can be applied to the vertex of the actuating member of the moving platform, and then the function generator provided by ADAMS/VIEW is used to define a function about time to construct the motion excitation, which guides the vertices of the actuating member of the moving platform to realize the motion of the expected trajectory, which saves the cumbersome process of inverse solving equations, and improves the efficiency of the solving process. The results obtained in this way are the same as those analyzed using inverse kinematics, and it is much easier to use the method in the paper, which allows different motions to be realized just by changing the equations of the trajectory of the end. For this paper is to study the influence of the error sources of this hybrid robot on the robot's end situation, the expected results can be achieved by using the method in the text, so the method in the text is used.

  1. The Inverse kinematics model is missing in the paper!

Author's Response: We sincerely appreciate the valuable comments. Kinematic analysis of the robot was added to the changed article, adding inverse kinematics modeling to make the article more complete. The added inverse kinematics model is in the revised draft's 4.1.1 Summary, page 6, line 225.

  1. What are the requirements of the intended application in terms of accuracy?

Author's Response: Thank you very much for your comment. Because our robot is still in the design stage, the current expected application requirement is to be able to surface machining requirements, to meet the current machining accuracy, in the future research will continue to improve the accuracy of the robot along the direction of research, so that the design of the robot can achieve higher accuracy requirements. This is also described on page 22, line 682 of the revised draft.

  1. How can the carried out work help designing the presented robot?

Author's Response: We sincerely appreciate the valuable comments. The research in this paper is to conduct an error analysis of the robot to find out the most significant factors affecting the end of the robot by analyzing the sources of error, modeling, and simulation experiments. Based on this paper, when designing this robot one should focus on the effect of these significant factors on the robot's end, and strictly control these sources of error during the machining, manufacturing and assembly process. There is a related description in the Discussion section of the revised draft, on page 22, line 679.

We sincerely thank the reviewers once again for your comments, and all of the comments made have been revised.